# Natural Language Decompositions of Implicit Content Enable Better Text Representations

**Alexander Hoyle**[*]      **Rupak Sarkar**[*]      **Pranav Goel**      **Philip Resnik**

Computer Science      Computer Science      Computer Science      UMIACS, Linguistics

University of Maryland

{hoyle,rupak,pgoel1}@cs.umd.edu, resnik@umd.edu

## Abstract

When people interpret text, they rely on inferences that go beyond the observed language itself. Inspired by this observation, we introduce a method for the analysis of text that takes implicitly communicated content explicitly into account. We use a large language model to produce sets of propositions that are inferentially related to the text that has been observed, then validate the plausibility of the generated content via human judgments. Incorporating these explicit representations of implicit content proves useful in multiple problem settings that involve the human interpretation of utterances: assessing the similarity of arguments, making sense of a body of opinion data, and modeling legislative behavior. Our results suggest that modeling the meanings behind observed language, rather than the literal text alone, is a valuable direction for NLP and particularly its applications to social science.[1]

## 1 Introduction

The meaning and import of an utterance are often underdetermined by the utterance itself. Human interpretation involves making inferences based on the utterance to understand what it communicates (Bach, 1994; Hobbs et al., 1993). For the disciplines and applications that are concerned with making sense of large amounts of text data, human interpretation of each individual utterance is intractable. Some NLP methods are designed to facilitate human interpretation of text by aggregating lexical data; for example, dictionaries map words to constructs (e.g., Pennebaker et al., 2001), and topic models discover interpretable categories in a form of automated qualitative content analysis (Grimmer and Stewart, 2013; Hoyle et al., 2022). Much of the time, though, these techniques operate over surface forms alone, limiting their ability

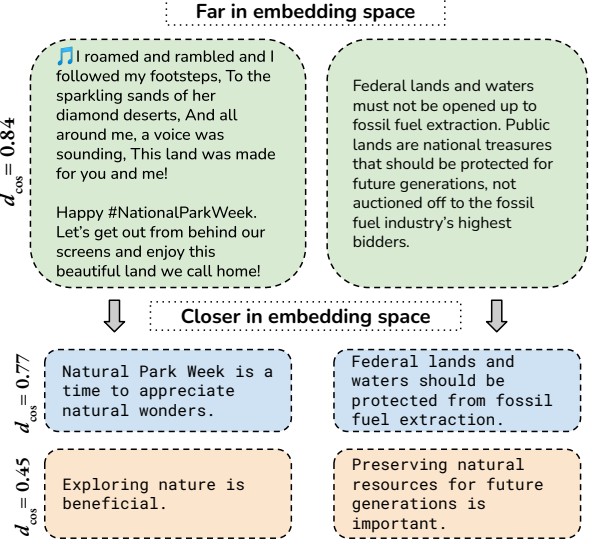

Figure 1: Example showing a pair of tweets from legislators along with inferentially-related propositions. Embeddings over observed tweets have a high cosine distance, while embeddings over (different types of) propositions place them closer to each other (see § 5).

to capture implicit content that facilitates human interpretation. While contextual embeddings are a step in this direction, these representations remain dominated by lexical content (Zhang et al., 2019).

In this work, we introduce a framework for the interpretation of text data at scale that takes implicitly communicated content more explicitly into account. Specifically, we generate sets of straightforward propositions that are inferentially related to the original texts. We refer to these as *inferential decompositions* because they break the interpretations of utterances into smaller units. Broadly speaking, we follow Bach (2004, p. 476) in distinguishing "information encoded in what is uttered" from extralinguistic information. Rather than being logical entailments, these generations are *plausible entailments* of the kind found in discussions of textual entailment (Dagan et al., 2009) and implicature (Davis, 2019).This idea relates to decompositional semantics (White et al., 2020), but eschews

---

[*] Equal contribution.

[1] Code and data available at github.com/ahoho/inferential-decompositions

linguistically-motivated annotations in favor of a more open-ended structure that facilitates direct interpretation in downstream applications.

We perform this process using a large language model (LLM), specifying a practitioner protocol for crafting exemplars that capture explicit and implicit propositions based on utterances sampled from the corpus of interest, thereby guiding the language model to do the same (§ 2). In designing our approach, we observe that the inherent sparseness of text data often makes it useful to represent text using lower-dimensional representations. Accordingly, our notion of decomposition (and the generation process) encourages propositions that contain simple language, making them both easier to interpret and more amenable to standard embedding techniques. We observe that the viability of this technique rests on models' ability to reliably generate real-world associations (Petroni et al., 2019; Jiang et al., 2020; Patel and Pavlick, 2022), as well as their capacity to follow instructions and mirror the linguistic patterns of provided exemplars (Brown et al., 2020; Liu et al., 2022).

We situate our approach within the *text-as-data* framework: the "systematic analysis of large-scale text collections" that can help address substantive problems within a particular discipline (Grimmer and Stewart, 2013). First, we validate our method with human annotations, verifying that generated decompositions are plausible and easy to read, and we also show that embeddings of these inferential decompositions can be used to improve correlation with human judgments of argument similarity (§ 3). We then turn to two illustrations of the technique's utility, both drawn from real-world, substantive problems in computational social science. The first problem involves making sense of the space of public opinion, facilitating human interpretation of a set of comments responding to the US Food and Drug Administration's plans to authorize a COVID-19 vaccine for children (§ 4). The second involves the question of how likely two legislators are to vote together based on their tweets (§ 5).

## 2 The Method and its Rationale

The key idea in our approach is to go beyond the observable text to explicitly represent and use the kinds of implicit content that people use when interpreting text in context.

Consider the sentence *Build the wall!*. Following Bender and Koller (2020), human interpreta-

```
Human utterances communicate propositions that
may or may not be explicit in the literal meaning of
the utterance. For each utterance, state the implicit
and explicit propositions communicated by that ut-
terance in a brief list. Implicit propositions may be
inferences about the subject of the utterance or about
the perspective of its author. All generated proposi-
tions should be short, independent, and written in
direct speech and simple sentences.
###
INPUT: { utterance }
OUTPUT: { inference 1 } | { inference 2 } | ...
```

Figure 2: A condensed version of our prompt to models.

tion of this sentence in context involves deriving the speaker's communicative intent $i$ from the expression itself, $e$, together with an implicit universe of propositions *outside* the utterance, $U$—world knowledge, hypotheses about the speaker's beliefs, and more. In this case, some elements of $U$ might be factual background knowledge such as "The U.S. shares a border with Mexico" that is not communicated by $e$ itself. Other elements might include implicitly communicated propositions such as "immigration should be limited". Propositions from this latter category, consisting of relevant plausible entailments *from* the utterance, we denote as $\mathcal{R} \subset U$.[2]

We are motivated by the idea that, if human interpretation includes the identification of plausible entailments $\mathcal{R}$ based on the expressed $e$, automated text analysis can also benefit from such inferences, particularly in scenarios where understanding text goes beyond "who did what to whom". The core of our approach is to take an expression and *explicitly* represent, as language, a body of propositions that are related inferentially to it.

Operationally, our method is as follows:

1. For the target dataset, randomly sample a small number of items (e.g., tweets).
2. Craft explicit and implicit propositions relevant to the items (inferential decompositions expressed as language) following the instructions in appendix A.2 to form exemplars.
3. Prompt a large language model with our instructions (fig. 2) and these exemplars.
4. Confirm that a random sample of the generated decompositions are plausible (§ 3).[3]
5. Use the decompositions in the target task.

---

[2]Cf. Bach's characterization of pragmatic information as "relevant to the hearer's determination of what the speaker is communicating ... generated by, or at least made relevant by, the act of uttering it". Similarly, interpretation as abduction (Hobbs et al., 1993) is an inferential process that starts *from* the utterance.

[3]While this step is not necessary, we recommend validating generations, particularly in sensitive use cases.

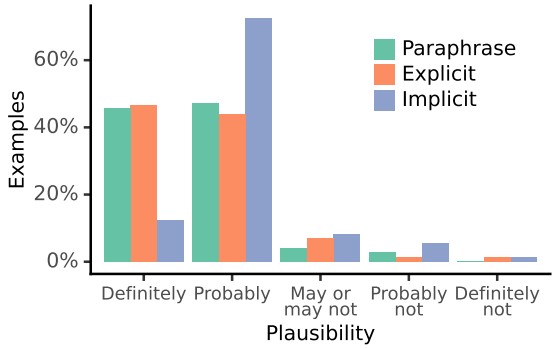

Figure 3: Human-judged plausibility of inferentially-related propositions for different inference types. The vast majority of inferred propositions are plausible.

| | | Decompositions | | |
|---|---|---|---|---|
| | Baseline | Explicit | Implicit | Both |
| *Implicit* | | | | |
| Arg Facet | 54.01 | 59.98 | 58.84 | 59.79 |
| BWS Arg. | 53.74 | 55.32 | 54.63 | 55.09 |
| UKP Aspect | 50.04 | 52.86 | 53.40 | 53.39 |
| *Explicit* | | | | |
| Twitter-STS | 69.05 | 72.69 | 69.92 | 71.92 |
| SICK-R | 80.59 | 81.39 | 75.92 | 80.00 |
| STS-B | 83.42 | 81.27 | 77.59 | 81.76 |

Table 1: Measured against human similarity judgments, similarities computed using inferential decompositions are generally better than similarities computed using the observable text (Spearman $\rho$). Improvements from the baseline are underlined.

To ground these propositions in a context, we develop domain-dependent user guidelines (step 2) for exemplar creation, limiting the focus to inferences about the utterance topic and its speaker (and explicit content, appendix A.2).[4]

Continuing with the example $e$ of *Build the wall!*, the method might generate the augmented representation $\mathcal{R} = \{$*A US border wall will reduce border crossings*, *Illegal immigation should be stopped*, ...$\}$ as plausible inferences about the speaker's perspective (an actual example appears in fig. 1).

Because the propositions are expressed in simple language, they are easier to represent with standard embedding techniques: when using $K$-means clustering to over the embedded representations, the clusters are far more readable and distinct than baselines (§ 4). Hence, this approach facilitates interpretation of text data at scale.[5]

## 3 Generation Validity

Text analysis methods used in the computational social sciences have known issues with validity. The interpretation of unsupervised models is particularly fraught—leading to potentially incorrect inferences about the data under study (Baden et al., 2021). We validate the two components of our approach: the quality of the generations and the similarity of their embeddings.

First, if we want to use the generated inferential decompositions in downstream applications, it is important that they are *reasonable* inferences from the original utterance. Large language models are known to hallucinate incorrect information

in some circumstances (Maynez et al., 2020; Cao et al., 2022; Ji et al., 2023), but can also exhibit high factuality relative to prior methods (Goyal et al., 2023). This leads to the question: *Do language models reliably produce plausible explicit & implicit propositions?*

Second, since similarity over embedded text underpins both the text analysis (§ 4) and downstream application (§ 5) that validate our approach, we measure the correlation of embedding and ground-truth similarities for several standard semantic-textual-similarity tasks. Assuming that human similarity judgments make use of both explicit and implicit inferences, we also ask *whether including such information in sentence representations improve automated estimates of similarity.*

**Generation of decompositions.** We generate inferential decompositions for datasets across a diverse set of domains. Our method should effectively encode stance, often an implicit property of text, so we adopt three argument similarity datasets (*Argument Facets* from Misra et al. 2016; *BWS* from Thakur et al. 2020-10; and *UKP* from Reimers et al. 2019). As a reference point, we also select several standard STS tasks from the Massive Text Embedding Benchmark, which are evaluated for their observed semantic similarity (MTEB, Muennighoff et al., 2022).[6] We also use the datasets underpinning our analyses in sections 4 and 5: public commentary on FDA authorization of COVID-19 vaccines, and tweets from US senators.

To generate decompositions, we use the instructions and exemplars in appendix A.4, further dividing the exemplars into *explicit* and *implicit*

---

[4]In principle, alternative guidelines meeting other desiderata are possible, and can be validated with the same process.

[5]"Scale" refers to dataset sizes that render human analysis too costly, and is constrained only by computational budget.

[6]For the Twitter dataset (Xu et al., 2015), we use the original 5-scale Likert similarity, not the binary scores in MTEB.

categories, as determined by our guidelines (appendix A.2). See fig. 1 for an illustration. For the human annotation, we sample 15 examples each from STS-B (Cer et al., 2017), BWS, Twitter-STS (Xu et al., 2015), and our two analysis datasets. The language model is text-davinci-003 (INSTRUCTGPT, Ouyang et al., 2022), and embeddings use all-mpnet-base-v2 (Reimers and Gurevych, 2019). Throughout this work, we use nucleus sampling with $p = 0.95$ (Holtzman et al., 2019) and a temperature of 1.

**Human annotation of plausiblity.** In answering the first question, a set of 80 crowdworkers annotated the extent to which a decomposition is reasonable given an utterance—from "1 - Definitely" to "5 - Definitely not"—and whether it adds new information to that utterance (full instructions in appendix A.5.1, recruitment details in appendix A.5). We majority-code both answers, breaking ties for the plausibility scores with the rounded mean.

In the vast majority of cases (85%-93%), the generated decompositions are at least "probably" reasonable (fig. 3). As expected, the plausibility of *implicit* inferences tends to be less definite than either a paraphrase baseline or explicit inferences—but this also speaks to their utility, as they convey additional information not in the text. Indeed, implicit inferences add new information 40% of the time, compared to 7% for explicit inferences (and 15% for paraphrases). As further validation to support the analyses in § 5, a professor of political science annotated the implicit decompositions of observable legislative tweets, finding 12 of 15 to be at least "probably reasonable," two ambiguous, and one "probably not reasonable".

**Semantic Textual Similarity.** Here, we measure whether our method can improve automated measurements of semantic sextual similarity. For each example in each of the STS datasets, we form a set $S_i = \{s_i, \tilde{s}_{i,1}, \tilde{s}_{i,2}, \ldots, \tilde{s}_{i,n}\}$ consisting of the original utterance and $n$ decompositions. As baseline, we computed the cosine similarity comparisons between embeddings of the original sentences $s_i, s_j$, obtained using all-mpnet-base-v2. Pairwise comparisons for expanded representations $S_i, S_j$, were scored by concatenating the embedding for $s_i$ with the mean of the embeddings for the $s_{i,*}$.[7]

Our method substantially improves correlation

| $K$ | Method | Silhouette↑ | CH↑ | DB↓ |
|---|---|---|---|---|
| 15 | Comments | 0.052 | 247 | 3.41 |
| | Sentences | 0.042 | 219 | 3.74 |
| | Decompositions (ours) | **0.090** | **329** | **3.03** |
| 25 | Comments | 0.035 | 172 | 3.28 |
| | Sentences | 0.035 | 152 | 3.64 |
| | Decompositions (ours) | **0.096** | **239** | **2.80** |
| 50 | Comments | 0.029 | 104 | 3.26 |
| | Sentences | 0.042 | 93 | 3.51 |
| | Decompositions (ours) | **0.114** | **153** | **2.73** |

Table 2: Intrinsic metrics of clustering quality. On a random subsample of 10k comments, sentences, and decompositions, the intrinsic metrics rank our model higher both for a fixed number of clusters (bolded) and across clusters (underlined). CH is the Calinski-Harabasz Index and DB is Davies-Bouldin.

on the argument similarity datasets over the embedding baseline (table 1), where pairs are annotated for the similarity of their position and explanation with respect to a particular topic (e.g., supporting a minimum wage increase by invoking inflation). In this task, explicit decompositions resemble the implicit ones—annotators give similar proportions of "probably reasonable" scores to both types.[8]

On the conversational Twitter-STS dataset, the method also shows improvement, likely due to the colloquial and contextualized nature of the original utterances.[9] Unsurprisingly, on standard STS benchmarks, the implicit method fares worse, likely because it over-generalizes from specific instances that reduce precision, even if they remain correct: "A person is mixing a pot of rice." → "The person is preparing food." Indeed, our method is *intended* to create such generalizations to assist in interpretability at scale, not to improve STS tasks.[10]

## 4 Inferential Decompositions Help Theme Discovery

Since the augmented representations we are creating go beyond the observed text to inferentially related propositions, we expect it to be useful in problem settings where observable text is the "tip of the iceberg" — intuitively, problems where it is particularly important to consider not only what was said, but what is *behind* what was said. Specifically,

---

[7]Approaches designed to measure the similarity of sets of vectors gave similar results(e.g., Zhelezniak et al., 2020)

[8]For example, from a 38-word utterance, *explicit*: "The minimum wage should be higher than $7.25"; *implicit*: "The current minimum wage is insufficient"

[9]Even the "explicit" setting generates "Chris Kelly has died" from the original "RIP To tha Mac daddy Chris Kelly".

[10]Altering the prompt to support STS by instead generating paraphrases leads to state-of-the-art results, appendix A.3.

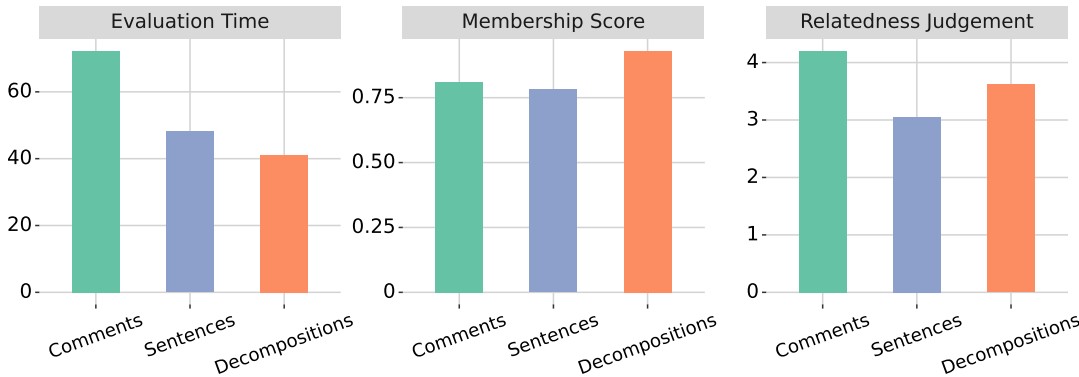

Figure 4: Human evaluation of clustering outputs. Clusters of decompositions (our method) take significantly less time to review and are more distinctive from one another. Relatedness scores are high for the observed comments, but significantly worse membership identification scores reveal this to be a spurious result owed to the topical homogeneity of the dataset (all comments are about COVID vaccines). All differences are significant at $p < 0.05$ except membership scores between comments and sentences and evaluation times for sentences and decompositions.

we ask *whether the representation of comments' explicit & implicit propositions lead to improved discovery of themes in a corpus of public opinion.* Understanding public opinion on a contentious issue fits that description: expressions of opinion are generated from a more complex interplay of personal values and background beliefs about the world. This is a substantive real-world problem; in the US, federal agencies are required to solicit and review public comments to inform policy.

Our approach is related to efforts showing that intermediate text representations are useful for interpretive work in the computational social sciences and digital humanities, where they can be aggregated to help uncover high-level themes and narratives in text collections (Bamman and Smith, 2015; Ash et al., 2022). In a similar vein, we cluster inferential decompositions of utterances that express opinion to uncover latent structure analogous to the discovery of narratives in prior work. We analyze a corpus of public comments to the U.S Food and Drug Administration (FDA) concerning the emergency authorization of COVID-19 vaccinations in children — in terms of content and goals, our application resembles the latent argument extraction of Pacheco et al. (2022), who, building on content analysis by Wawrzuta et al. (2021), clustered tweets relating to COVID-19 to facilitate effective annotation by a group of human experts. In our case, we not only discover valuable latent categories, but we are able to assign naturalistic category labels automatically in an unsupervised way.[11]

---

[11]In preliminary experiments, topic model outputs were of mixed quality.

**Dataset.** We randomly sampled 10k responses from a set of about 130k responses to a request for comments by the FDA regarding child vaccine authorization.[12] Our dataset contains often-lengthy comments expressing overlapping opinions, colloquial language, false beliefs or assumptions about the content or efficacy of the vaccine, and a general attitude of vaccine hesitancy (see the "Comment" column of table 10 for examples).

**Method.** We generate 27,848 unique inferential decompositions from the observable comments at an average of 2.7 per comment. We use $K$-means clustering to identify categories of opinion, varying $K$. Specifically, two authors created 31 exemplars from seven original comments from the dataset that exhibit a mixture of implicit proposition types (table 10). In addition to clustering the observed comments themselves as a baseline, as a second baseline we split each comment into its overt component sentences and cluster the full set of sentences. This results in 10k comments, 45k sentences, and 27k decompositions.

**Automated Evaluation.** We lack ground truth labels for which documents belong to which cluster, so we first turn to intrinsic metrics of cluster evaluation: the silhouette (Rousseeuw, 1987),

---

[12]regulations.gov/document/FDA-2021-N-1088-0001. Comments are public and users can elect to post anonymously. We obtained permission from the agency to use these data. We will not directly release the data out of caution, because the original authors did not explicitly consent to redistribution, but we refer interested researchers to https://www.regulations.gov/bulkdownload and Pampell (2022). Note that some comments can contain upsetting language, which we communicated to annotators.

| Source | Cluster 1 | Cluster 2 | Cluster 3 |
|---|---|---|---|
| Decomposition Clusters | The vaccine may be harmful to children. Children are vulnerable to the long-term effects of the vaccine. Vaccines are still under trials and their side effects are unknown. The vaccine has not been properly tested. There is not enough data on the vaccine's long-term effects. | Natural immunity is better than vaccine-induced immunity. Natural immunity is important for children to develop. Natural immunity is more effective. Natural immunity is better for fighting the covid virus. People should develop immunities to stay healthy. Natural immunity should be reflected in science. | The government is attempting to control citizens. American rights are being eroded. The government should investigate the use of We need to protect our children. The US values freedom and choice. |
| Expert Narratives Wawrzuta et al. (2021) | The vaccine is not properly tested, it was developed too quickly | Natural methods of protection are better than the vaccine | Lack of trust in the government |
| Crowdworker Label | Long-term vaccine worries | Natural Immunity | Control by government |

Table 3: For public commentary datasets about COVID-19, clusters of inferential decompositions (our approach, top row) align with arguments discovered independently by Wawrzuta et al. (2021) (middle row). The overlap is strong despite the commentary coming from different platforms (Government website & Facebook) and countries (US & Poland). In addition, outside of the exemplars passed to the LLM (table 10), our approach is also entirely unsupervised. In the bottom row, we show an illustrative label for each cluster from a crowdworker.

Calinski-Harabasz (Caliński and Harabasz, 1974), and Davies-Bouldin (Davies and Bouldin, 1979) scores; roughly speaking, these variously measure the compactness and distinctiveness of clusters. Since metrics can be sensitive to the quantity of data in a corpus (even if operating over the same content), we subsample the sentence and decomposition sets to have the same size as the comments (10k).[13] Clusters of decompositions dramatically outperform clusters of comments and sentences across all metrics for each cluster size—in fact, independent of cluster size, the best scores are obtained by decomposition clusters (Table 2).

**Human Evaluation.** Performance on intrinsic metrics does not necessarily translate to usefulness, so we also evaluate the cluster quality with a human evaluation. After visual inspection, we set $K = 15$. For a given cluster, we show an annotator four related documents and ask for a free-text label describing the cluster and a 1–5 scale on perceived "relatedness". We further perform a membership identification task: an annotator is shown an unrelated distractor and a held-out document from the cluster, and asked to select the document that "best fits" the original set of four. Participant information and other survey details are in appendix A.5.

Results are shown in Fig 4. While comment clusters receive a higher relatedness score, this is likely due to the inherent topical coherence of the dataset: there are often several elements of similarity between *any* two comments. A lower score in the membership identification task, however, indicates

that comment clusters are less distinct. Moreover, the comprehension time for comments is significantly longer than for sentences and decompositions (Evaluation Time in Fig 4), taking over 50% longer to read. On the other hand, clusters of decompositions strike a balance: they obtain moderately strong relatedness scores, can be understood the quickest, and are highly distinct.

**Convergent Validity.** Although further exploration is necessary, we find that our crowdworker-provided labels can uncover themes discovered from classical expert content analysis (table 3). For example, two crowdworkers assign labels containing the text "natural immunity" to the cluster in table 3—this aligns with the theme NATURAL IMMUNITY IS EFFECTIVE discovered in Pacheco et al. (2023) (through a process requiring more human effort) and a similar narrative in Wawrzuta et al. (2021). Meanwhile, this concept does *not* appear anywhere in the crowdworker labels for the baseline clusters of sentences or comments.

## 5 Decompositions support analyses of legislator behavior

Having established our method's ability to facilitate human interpretation of text data (§ 3 and § 4), we now examine the usefulness of generated inferential decompositions in a very different downstream application where the relevant analysis of text is likely to involve more than just overt language content. Here, we model legislator behavior using their speech (here, tweets), asking: *does the similarity between legislators' propositions help explain the similarity of their voting behavior?*

---

[13]Results are similar for the silhouette and Davies-Bouldin scores without subsampling; the Calinski-Harabasz is better for the sentences.

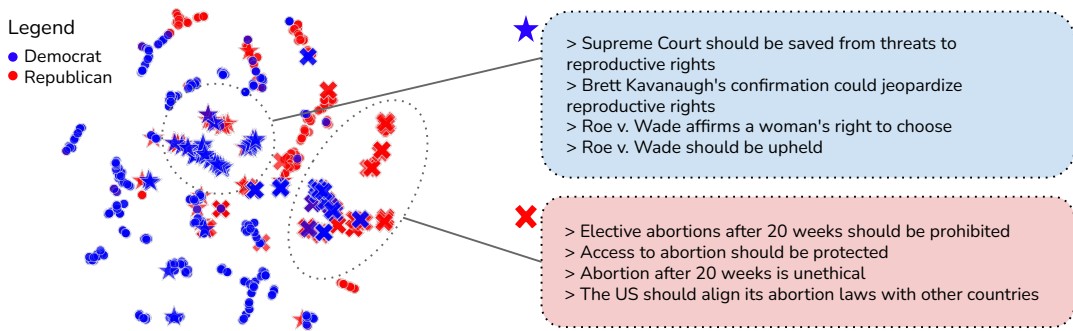

Figure 5: t-SNE (van der Maaten and Hinton, 2008) visualization of the embedding space of implicit inferred decompositions found in the "Abortion" topic from legislative tweets. ★ and ✖ are two clusters selected from 10 clusters obtained using $K$-means; ★ (59% Democrat) talks about the role of judiciary in reproductive rights, while the ✖ (73% Republican) talks about banning late stage abortion. Our method leads to more compact (better Silhouette, CH, and DB scores compared to tweets) and easier to interpret clusters that help with narrative discovery.

Traditional theories of homophily (McPherson et al., 2001) suggest that shared properties (e.g., electoral geography, Clark and Caro, 2013) increase the likelihood that two legislators vote the same way — and many research questions have centered around such *co-voting* (Ringe et al., 2013; Peng et al., 2016; Wojcik, 2018). In preliminary experiments we found when modeling co-voting, text similarity between legislator was a valuable predictor of co-voting (Goel, 2023). Here we posit that such modeling can further benefit by going beyond surface text to capture inferentially related propositions related to viewpoints.

Consider Figure 1. While the observable tweets themselves are not similar in their surface form, they express similar views regarding the importance of the preservation of nature, something that would be clear to a human reader interpreting them. This can be captured by inferential decompositions that reveal authors' viewpoints toward the issue.

We operationalize our method for this purpose by creating exemplars that contain inferences about the utterance topic and perspective (table 12 in appendix). This guides the language model toward domain-relevant facets of similarity between the texts, and therefore the authors of the texts, that may not be apparent from the surface form.

**Model Setup.** For the task of modeling co-vote behavior, we extend the framework introduced by Ringe et al. (2013) to incorporate individuals' *language* into the model. At a high level, we operationalize legislator homophily by measuring the similarities of their embedded speech Twitter. Following Ringe et al. (2013) and Wojcik (2018), we model the log odds ratio of the *co-voting rate* be-

tween a pair of legislators $i, j$ using a mixed effects regression model, controlling for the random effects of both actors under consideration. The co-vote rate $\lambda$ is the number of times the legislators vote the same way — yea or nay — divided by their total votes in common within a legislative session.

$$\mathbb{E}\left[\log\left(\frac{\lambda_{ij}}{1 - \lambda_{ij}}\right)\right] = \beta_0 + \boldsymbol{\beta}^\top \boldsymbol{x}_{ij} + a_i + b_j \quad (1)$$

$\beta_*$ are regression cofficients, and $a_i, b_j$ model random effects for legislators $i, j$.

$\boldsymbol{x}_{ij}$ is an $n$-dimensional feature vector, where each element is a similarity score that captures a type of relationship between legislators $i$ and $j$. While these features have traditionally included state membership, party affiliation, or Twitter connections (Wojcik, 2018) or joint press releases (Desmarais et al., 2015); we consider the language similarity between pairs of legislators based on our proposed method.

**Dataset.** The goal is to represent each legislator using their language in such a way that we can measure their similarity to other legislators—our informal hypothesis of the data-generating process is that a latent ideology drives both vote and speech behavior. Specifically, we follow Vafa et al. (2020) by using their tweets; data span the $115^{th} - 117^{th}$ sessions of the US Senate (2017–2021).[14]

We further suppose that ideological differences are most evident when conditioned on a particular issue, such as "the environment"; in fact, Bateman et al. (2017) note that aggregated measures like ideal points mask important variation across issues. To this end, we first train a topic

---

[14]Tweets from github.com/alexlitel/congresstweets, votes from voteview.com/about.

model on Twitter data to group legislator utterances into broad issues.[15] Two authors independently labeled the topics they deemed most indicative of ideology, based on the top words and documents from the topic-word ($\boldsymbol{\beta}^{(k)}$) and topic-document ($\boldsymbol{\theta}^{(k)}$) distributions.[16] Then, for each selected topic $k$ and legislator $l$, we select the top five tweets $U_l^{(k)} = \{u_{l,1}^{(k)}, \ldots u_{l,5}^{(k)}\}$, filtering out those with estimated low probability for the topic, $\theta_{ul}^{(k)} < 0.5$. Finally, we generate a flexible number of inferential decompositions for the tweets $R_l^{(k)}$ using GPT-3.5.[17] We use two prompts each containing six different exemplars to increase the diversity of inferences. The collections of tweets and decompositions are then embedded with Sentence-Transformers (`all-mpnet-base-v2` from Reimers and Gurevych, 2019).

**Language Similarity.** To form a text-based similarity measure between legislators $i$ and $j$, we first compute the pairwise cosine similarity between the two legislators' sets of text embeddings, $s_{\cos}\left(U_i^{(k)} \times U_j^{(k)}\right)$ (or $R$ for the decompositions). The pairs of similarities must be further aggregated to a single per-topic measure; we find the $10^{\text{th}}$ percentile works well in practice (taking a maximum similarity can understate differences, whereas the mean or median overstate them, likely due to finite sample bias; see Demszky et al. 2019). This process creates two sets of per-topic similarities for each $(i, j)$-legislator pair, $s_{ij}^{(k)}(u), s_{ij}^{(k)}(r)$.

**Results.** Similarities based on inferential decompositions, $s_{ij}(r)$, both help explain the variance in co-vote decisions as well as help predict co-voting agreement over and above similarities based on the observed utterances (tweets) alone, $s_{ij}(u)$ (table 4).[18] Specifically, for the regression models, across three Senate sessions, the mixed effects model that uses similarity in decompositions ($R$;

[15]Text was processed using the toolkit from Hoyle et al. (2021), and modeled with collapsed Gibbs-LDA (Griffiths and Steyvers, 2004) implemented in MALLET (McCallum, 2002).

[16]E.g, top words "border, crisis, biden, immigration" correspond to the politically-charged issue of immigration; the more benign "tune, live, watch, discuss" covers tweets advertising a media appearance to followers. Both annotators initially agreed on 92% of labels; disagreements were resolved via discussion. The final set of 33 topic-words can be found in github.com/ahoho/inferential-decompositions.

[17]Results in table 4 are extremely similar when using Alpaca-7B, suggesting the findings are robust.

[18]We use membership in the same political party as a control variable in each of the mixed effects models.

| | $115^{th}$ | $116^{th}$ | | $117^{th}$ | |
| --- | --- | --- | --- | --- | --- |
| | $\hat{\boldsymbol{\beta}}\uparrow$ | $\hat{\boldsymbol{\beta}}\uparrow$ | **MAE**↓ | $\hat{\boldsymbol{\beta}}\uparrow$ | **MAE**↓ |
| Sim. $U$ | 10.64 | 11.86 | 744.81 | 17.36 | 981.56 |
| Sim. $R$ | 12.69 | 17.31 | 736.21 | 23.0 | 964.86 |
| Sim. $U$ | 6.02 | 4.40 | 741.59 | 11.09 | 935.45 |
| + Sim. $R$ | 7.01 | 12.67 | | 10.82 | |

Table 4: Explanatory power ($\hat{\beta}$) and predictive capacity (**MAE**) of text similarity measures in a mixed effects model of co-vote behavior for 3 sessions of the US Senate. Coefficients are large and significantly above zero ($p < 0.001$). For the mean absolute error (MAE), we multiply the predicted co-vote agreement error with the total number of votes in common for the legislator pair.

row 2 in table 4) has a higher regression coefficient ($\hat{\beta}$) than the corresponding coefficient for similarity in utterances ($U$; row 1). This also holds for a model that uses both similarity measures in the regression (row 3).

Additionally, we compare the predictive capacity of the two similarity measurements in two scenarios: we train the model on data from the $115^{th}$ Senate to predict co-vote for the $116^{th}$, and train on data from the $115^{th}$ and $116^{th}$ sessions to predict co-vote for the $117^{th}$. Using similarity in decompositions leads to a lower MAE between predicted and actual co-vote agreement than using similarity in utterances; both similarity measures together further reduce the error for the $117^{th}$ Senate (table 4).

Examples of utterances and their decompositions in fig. 6 help contextualize these results. For the left-hand example, the method infers a shared implicit proposition—"President Trump is weak"—that underlies two tweets with little *observed* text in common. However, the method can occasionally overstate similarities between utterances (and thus, between legislators): while the decompositions in the right-hand example are valid inferences, they are also *overly general* ("voting is important").

Our approach further uncovers the narratives that emerge among legislators from different parties around a particular issue. In discussions around abortion and reproductive health, our decompositions capture fine-grained viewpoints about the role of supreme court and judiciary, and the contentious debate around late stage abortions (Fig. 5). Clustering over implicit decompositions reveal finer opinion-spaces that succinctly capture author viewpoints towards facets of a particular issue.

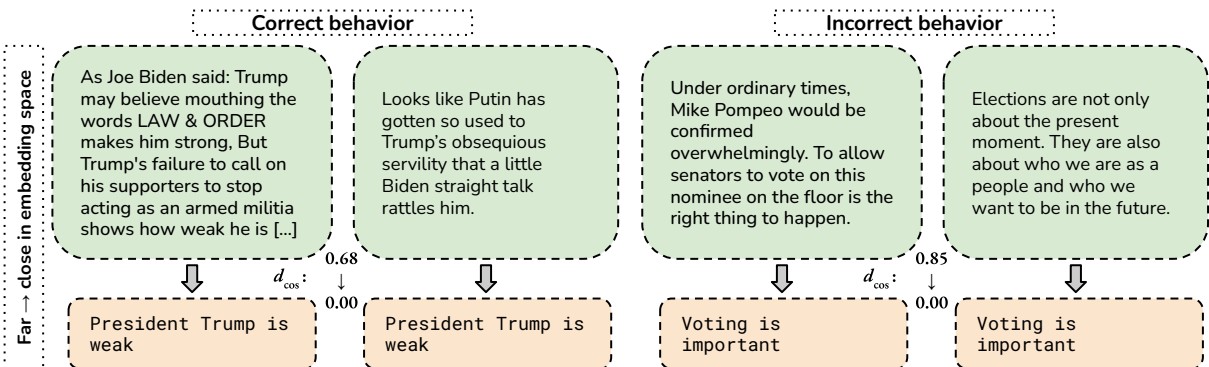

Figure 6: Pairs of legislative tweets (green) and associated decompositions (orange). Here, we show instances where embeddings of the decompositions are *closer* than embeddings of the original tweets. The example on the left shows the method working as intended, whereas the example on the right is undesired behavior. Although the method generates multiple decompositions per tweet, we only show the two closest. In appendix A.1, we discuss instances where the decompositions are *more distant* than the tweets.

## 6 Related Work

As a computational text analysis method that operates over reduced representations of text data, our work is similar to that of Ash et al. (2022) and Bamman and Smith (2015). Closest to our clustering effort (§ 4), Ernst et al. (2022) extract and cluster paraphrastic propositions to generate summaries. However, they constrain themselves to the observed content alone, limiting their ability to capture context-dependent meaning.

Our method continues the relaxation of formal semantic representations in NLP, a process exemplified by the Decompositional Semantics Initiative (White et al., 2016), which aims to create an intuitive semantic annotation framework robust to variations in naturalistic text data. The generated inferential decompositions in our work are human-readable and open-ended, structured by a user-defined schema in the form of exemplars. In this way, we also follow a path forged by natural language inference (Bowman et al., 2015; Chen et al., 2017), which has increasingly relaxed semantic formalisms when creating datasets. In future work, we plan to relate both the exemplars and outputs to formal categories and investigate their utility in downstream tasks.

Our methods for generating decompositions are distinct from *extracting* commonsense knowledge from text—e.g., through templates (Tandon et al., 2014), free-form generation from LLMs Bosselut et al. (2019), or script induction (Schank and Abelson, 1975) (which can also use LLMs, Sancheti and Rudinger 2022). Our decompositions are not designed to produce general commonsense knowledge such as "rain makes roads slippery" (Sap et al.,

2020), but instead to surface the explicit or implicit propositions inferrable from an utterance.

Our work also bears a similarity to Opitz and Frank (2022), who increase the interpretability of sentence embeddings using an AMR graph for a sentence. However, AMR graphs are also tied to the information present in an utterance's surface form (and, moreover, it is unclear whether AMR parsers can accommodate noisier, naturalistic text).

LLMs have been used to generate new information ad-hoc in other settings, for example by augmenting queries (Mao et al., 2020) or creating additional subquestions in question-answering (Chen et al., 2022). In contemporaneous work, Ravfogel et al. (2023) use an LLM to generate abstract descriptions of text to facilitate retrieval. In Gabriel et al. (2022), the authors model writer intent from a headline with LLMs. Becker et al. (2021) generate implicit knowledge that conceptually connects contiguous sentences in a longer body of text.

## 7 Conclusion

Our method of inferential decompositions is useful for text-as-data applications. First, we uncover high-level narratives in public commentary, which are often not expressed in surface forms. Second, we show that, by considering alternative representations of legislators' speech, we can better explain their joint voting behavior. More broadly, treating implicit content as a first-class citizen in NLP—a capability enabled via generation in large language models—has the potential to transform the way we approach problems that depend on understanding what is behind people's utterances, rather than just the content of the utterances themselves.

## 8 Limitations

Our validity checks in Section 3 reveal that while most decompositions are deemed reasonable by humans, some are not (Fig. 3.). It remains to be studied the extent to which implausible generations are affecting with the results, or if they are introducing harmful propositions not present in the original text. In future work, we will explore whether known political biases of language models (Santurkar et al., 2023) affect our results. Although an open-source model (specifically Alpaca-7B Taori et al. 2023) produces similar results to those reported, our main experiments primarily use models released by OpenAI, which may lead to potential reproducibility issues. All our analyses and experiments focus on utterances in the English language, which could limit the generalizability of our method. Relatedly, our experiments are also specific to the US sociocultural context and rely on models that are known to be Western-centric (Palta and Rudinger, 2023).

The embeddings could be made more sensitive to the particular use case. In future work, we plan to additionally fine-tune the embeddings so that they are more sensitive to the particular use case (e.g., establishing argument similarity, Behrendt and Harmeling, 2021)).

## 9 Ethics Statement

The work is in line with the ACL Ethics Policy. Models, datasets, and evaluation methodologies used are detailed throughout the text and appendix. The human evaluation protocol was approved by an institutional review board. No identifying information about participants was retained and they provided their informed consent. We paid per survey based on estimated completion times to be above the local minimum wage (appendix A.5). Participants were paid even if they failed attention checks. All the datasets were used with the appropriate or requested access as required. We acknowledge that we are using large language models, which are susceptible to generating potentially harmful content.

Generally, the potential for misuse of this method is not greater than that of the large language models used to support it. In theory, it is possible that a practitioner could draw incorrect conclusions about underlying data if the language model produces a large number of incorrect statements. To the extent that those conclusions inform downstream decisions, there could be a potential for negative outcomes. For this reason, we advocate for manual verification of a sample of outputs in sensitive contexts (step 4 of our protocol § 2).

## 10 Acknowledgements

We thank Sweta Agrawal for her abundant support and feedback on earlier drafts and Justin Pottle for assistance with data quality annotation. We also appreciate productive discussions with both Rachel Rudinger and Maharshi Gor, Jordan Boyd-Graber and Chenglei Si gave excellent notes during our internal paper clinic, and we thank our anonymous reviewers for their very helpful comments. This work was supported in part by the National Science Foundation under award #2008761, the Food and Drug Administration (FDA) of the U.S. Department of Health and Human Services (HHS) as part of a financial assistance award (Center of Excellence in Regulatory Science and Innovation cooperative agreement U01FD005946 with the University of Maryland), and a grant from the University of Maryland Social Data Science Center. Any opinions, findings, conclusions, or recommendations expressed in this material are those of the authors and do not necessarily represent the official views of, nor an endorsement, by any sponsor or by the U.S. government.

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

# A  Appendix

## A.1  Further Qualitative Analysis of Legislative Tweets

In Section 5, we showed that our method can place utterances with distant observed content closely through decompositions. In this section, we provide further illustrative examples where *small* utterance distances yield *larger* decomposition distances (Fig 7). As before, we show examples of "correct" behavior (where the method works as intended) as well as failures (where it does not).

In the left column of Fig 7, a tweet pair that discusses a similar topic (Coronavirus) has relatively similar embeddings even though they communicate considerably different content; appropriately, the generated decompositions are more distant. At the same time, although the two utterances in the right communicate very similar content, the generated decompositions are nonetheless further in embedding space—a problem is exacerbated by the open referent "they". In future iterations of the method, we plan to regularize outputs to avoid such issues.

## A.2  Decomposition Exemplar Creation Meta-Prompt

Below we present a condensed version of *human* instructions for the structured creation of exemplars.

Human utterances communicate propositions that may or may not be explicit in the literal meaning of the utterance. Your goal is to make a brief list of propositions that are implicitly or explicitly conveyed by the meaning of an utterance.

All the propositions you include should be short, independent, and written in direct speech and simple sentences. If possible, try to keep propositions to a single clause consisting of a subject, a predicate, and an object (don't worry too much about sticking to this format: noun phrases and prepositional phrases are acceptable). It may be helpful to "break up" propositions as necessary, and you should disambiguate unclear referents when possible ("vaccine" → "COVID vaccine").

By implicitly conveyed propositions, we mean propositions that are plausibly or reasonably inferred, even if they were not necessarily intended to be conveyed by the utterance. Shorter utterances will typically communicate fewer propositions. Longer texts may communicate several; we ask you to pre-

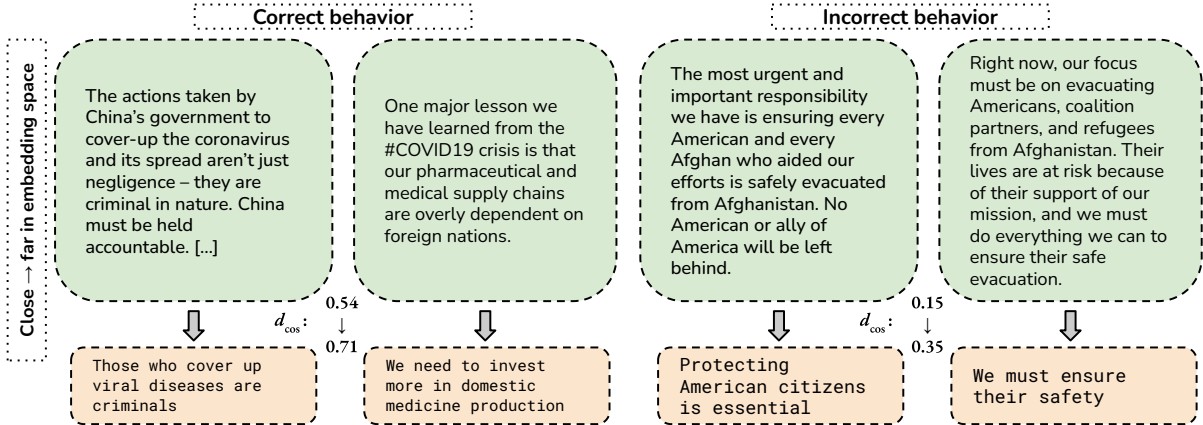

Figure 7: Pairs of legislative tweets (green) and associate decompositions (orange). Mirroring Fig fig. 6, we show instances where embeddings of the decomposition are *farther* than embeddings of the original tweets. The example on the left shows the method working as intended, whereas the example on the right shows undesired behaviour. Although the method generates multiple decompositions per tweet, we only show the two closest. We use the Sentence-Transformer model `all-mpnet-base-v2` to compute embeddings; generations are from Alpaca-7B.

fer writing propositions that are most central to the meaning that's being communicated. With the above in mind, limit yourself to five propositions per utterance. Write diverse propositions that are minimally redundant with one another. Propositions can fall into several categories.

**Explicit propositions [required].** To generate propositions in this category, rephrase elements of the utterance's explicit meaning as one or more simple propositions communicated by the utterance. It may be the case that almost no changes are necessary, and you will merely write a paraphrase. Including world knowledge is acceptable.

**Inferences about utterance subject [optional].** Make inferences from the utterance about the subject it is talking about. These are nontrivial but commonsense implications that can be reasonably and directly inferred from the utterance.

**Inferences about utterance perspective [optional]** Often, the author of an utterance is, intentionally or not, conveying information about their perspectives or preferences. Write down inferences from the utterance that are consistent with the author's perspective (as you understand it). These propositions sometimes take the form of modals ("should", "must"), or are general statements that may

|  | Sentence-T5 | + Paraphrases | |
|  |  | Alpaca | GPT-3 |
| --- | --- | --- | --- |
| SICK-R | 79.98 | 81.46 | 80.49 |
| STS-B | 83.93 | 85.49 | 86.28 |
| STS12 | 79.02 | 76.97 | 79.18 |
| STS13 | 88.80 | 88.44 | 89.18 |
| STS14 | 84.33 | 84.18 | 85.36 |
| STS15 | 88.89 | 89.28 | 89.28 |
| STS16 | 85.31 | 85.15 | 85.23 |
| STS17 | 88.91 | 89.94 | 90.29 |

Table 5: Spearman's $\rho$ for STS benchmarks using a paraphrase-based variant of our method. *Sentence-T5* embeds texts in each pair with `sentence-t5-xl` (Ni et al., 2022), + *Paraphrases* concatenates averaged embeddings of additional paraphrases generated with zero-shot Alpaca-7B or GPT-3. Improvements over Sentence-T5 are underlined. This demonstrates that our method can consistently improve STS correlations of arbitrary baseline embeddings.

express value judgments (not necessarily true). These should be written as implied statements. Rather than mentioning the author explicitly by writing something like "the author/speaker believes/thinks/fears X", just write the X.

## A.3 Adapting the method for semantic textual similarity tasks

Our proposed method decomposes utterances into related propositions, which does not markedly improve standard STS tasks that typically rely on the explicit content in text (see lower half of table 1).

However, an alternative approach that instead generates multiple expressions of the same meaning *does* improve sentence embedding performance. A single sentence is only one way of expressing a meaning, and in many settings, there is value in considering alternative ways of communicating that same meaning. For example, the BLEU score for machine translation evaluation (Papineni et al., 2002) works more effectively with multiple reference translations (Madnani et al., 2007). Dreyer and Marcu (2012) take this observation a step further by using packed representations to encode exponentially large numbers of meaning-equivalent variations given an original sentence.

Here, we show that improvements in sentence representations can obtained by expanding a sentence's form with multiple text representations restating the same content. Specifically, we represent every sentence $s_i$ by a set $S_i = \{s_i, \tilde{s}_{i,1}, \tilde{s}_{i,2}, \ldots, \tilde{s}_{i,n}\}$ consisting of the original utterance and $n$ paraphrases. As baseline, we computed the cosine similarity comparisons between embeddings of the original sentences $s_i, s_j$, obtained with the state-of-the-art Sentence-T5 (Ni et al., 2022).[19] Pairwise comparisons for expanded representations $S_i, S_j$, were scored by concatenating the embedding for $s_i$ with the mean of the embeddings for the $s_{i,*}$, $\left[e(s_i); \sum_k^n e(\tilde{s}_{i,k})\right]$. Three paraphrases per input were generated with both a 7B-parameter Alpaca model (Taori et al., 2023) and the OpenAI `text-davinci-003` (derived from **?**) using a 0-shot prompt: "`Paraphrase the following text.\n###\n Text: {input}\n Paraphrase: {output}`"

Table 5 summarizes results on STS tasks from the Massive Text Embedding Benchmark (MTEB, Muennighoff et al., 2022). Our method improves over the Sentence-T5 alone in all but one instance.[20]

### A.4 Prompts and Exemplars

We present our prompts in table 6 and our exemplars in tables 7 to 12.

---

[19]For all experiments we use the model `sentence-t5-xl`; directionally similar results were observed for the lightweight `all-mpnet-base-v2`

[20]Given the modularity of our approach, we expect that for instances where we there is an absolute improvement over the Sentence-T5 baseline, substituting the state-of-the-art embedding model would further improve results.

### A.5 Survey Details

**Inferential Decomposition Annotation.** 80 fluent English speakers in the US and UK with at least a high school diploma (or equivalent) annotate a random sample of 15 utterance-decomposition pairs, recruited via Prolific (`prolific.co`). We paid 2.10 USD/survey, which take a median 10 minutes.

**Clustering Annotation.** We recruited 20 fluent English speakers in the US and UK with at least a high school diploma via Prolific. After instruction with two artificially high- and low-quality clusters to help calibrate scores, participants reviewed a random sample of ten clusters from the pool of 45. We paid 3.50 USD/survey, median completion time was 17 min.

Total compensation to annotators was 257 USD; we targeted 14 USD/hour.

#### A.5.1 Instructions and Examples Provided to Survey Participants for Human Annotation and Evaluation

We present the instructions and examples provided to human annotators or survey participants, in order to validate the quality of the generations (§ 3), in fig. 8 and fig. 9 respectively.

We present the instructions and examples provided to human annotators or survey participants, in order to evaluate the quality of clustering offered by our approach (§ 4), in fig. 10 and fig. 11 respectively.

| Dataset | Prompt | Exemplars Per Prompt |
|---------|--------|---------------------|
| STS Paraphrases | ```
Paraphrase the following text.
###
Text:  <input>
Paraphrase:  <output>
``` | 0 |
| FDA Comments | ```
Human utterances contain propositions that may or may not be
explicit in the literal meaning of the utterance.  Given an
utterance, state the propositions of that utterance in a brief
list.  All generated propositions should be short, independent,
and written in direct speech and simple sentences.  A proposition
consists of a subject, a verb, and an object.
These utterances come from a dataset of public comments on the
FDA website concerning the covid vaccine.
===
Utterance:  <input>
Propositions:  <output>
``` | 6 |
| Explicit | ```
Human utterances communicate propositions.  For each utterance,
state the explicit propositions communicated by that utterance
in a brief list.  All generated propositions should be short,
independent, and written in direct speech and simple sentences.
If possible, write propositions with a subject, verb, and
object.  <dataset_description>
###
``` | 7 |
| Implicit | ```
Human utterances communicate propositions that may not be
explicit in the literal meaning of the utterance.  For each
utterance, state the implicit propositions communicated by
that utterance in a brief list.  Implicit propositions may be
inferences about the subject of the utterance or about the
perspective of its author.  All generated propositions should
be short, independent, and written in direct speech and simple
sentences.  If possible, write propositions with a subject, verb,
and object.  <dataset_description>
###
``` | 7 |
| All | ```
Human utterances communicate propositions that may or may not
be explicit in the literal meaning of the utterance.  For
each utterance, state the implicit and explicit propositions
communicated by that utterance in a brief list.  Implicit
propositions may be inferences about the subject of the
utterance or about the perspective of its author.  All generated
propositions should be short, independent, and written in direct
speech and simple sentences.  If possible, write propositions
with a subject, verb, and object.  <dataset_description>
###
``` | 7 |

Table 6: Prompt templates used for obtaining the decompositions. The FDA Comments and their generations were used in 4, and the generations on legislative tweets were used in 5. We used six exemplars along with the prompts in both of these cases. When generating from Alpaca-7B, we alter these templates according to their format.[21]

| Source | Utterance | Propositions: Explicit [Required] | Inferences about Utterance Subject [Optional] |
|---|---|---|---|
| STS-12 | The new policy gives greatest weight to grades, test scores and a student's high school curriculum. | A new policy emphasizes academic qualifications | Academic qualifications were less important in previous policies |
| STS-B | the anti-defamation league took out full-page advertisments in swiss and international newspapers earlier in april 2008 accusing switzerland of funding terrorism through the deal. | The anti-defamation league accused Switzerland of funding terrorism in April 2008 | The anti-defamation league is an influential organization / Switzerland made a deal with an anti-semitic group |
| STS-B | A woman is squeezing a lemon. | A woman squeezes juice from a lemon | A woman is preparing a food or beverage |
| STS-B | Palestinians clash with security forces in W. Bank | Palestinian and Israeli forces clashed in the West Bank | Palestinian and Israeli forces are in conflict |
| STS-B | The baby is laughing and crawling. | The baby laughs and crawls | The baby is happy |
| STS-B | Thieves steal Channel swimmer's wheelchair | Thieves steal wheelchair / Someone who swam the English Channel has a wheelchair | Disabled people can swim long distances |
| STS-12 | Russian stocks fell after the arrest last Saturday of Mikhail Khodorkovsky, chief executive of Yukos Oil, on charges of fraud and tax evasion. | Mikhail Khordorkovky's arrest led to a fall in Russian stocks / Mikhail Khordorkovky is accused of fraud and tax evasion | Yukos Oil is important to the Russian economy |

Table 7: Exemplars for inferential decomposition for the source type of Newswire/image captions. We sample $n$ exemplars from this set to form a prompt, per Table 6.

| Source | Utterance | Propositions: Explicit [Required] | Inferences about Utterance Subject [Optional] | Inferences about Utterance Perspective [Optional] |
|---|---|---|---|---|
| AFS | [Topic: death penalty] the use of the death penalty is an act of revenge which cannot be undone, so should the state be put on trial for putting to death an innocent person? | The death penalty is a form of revenge; The death penalty is irreversible | The death penalty allows for wrongful deaths | The death penalty appeals to base instincts; The justice system is imperfect; The state should be held accountable when it does wrong |
| BWS | [Topic: abortion] Roe v. Wade, the Supreme Court case that declared abortion a constitutional right, was decided in January 1973. | Roe v Wade declared abortion a constitutional right; Roe v Wade was decided in January 1973 | The law protects the right to abortion | Abortion rights have been longstanding |
| SemEval16 | [Topic: atheism] How do they come up with this insanity? Oh wait, they believe in a Sky-God. #fyilive | People who believe in a god came up with something insane | | Religion is irrational; Religious people do illogical things |
| UKP Aspect | [Topic: cryptocurrency] The new initiative will allow merchants to accept NXT and other virtual currencies as payment on their online stores and easily exchange it for fiat money. | Online stores can accept and exchange NXT.; NXT is a virtual currency | Some virtual currencies can be converted into fiat money.; Virtual currencies can be useful. | |
| UKP Sentential Arg Mining | [Topic: minimum wage] After 90 consecutive days of employment or the employee reaches 20 years of age, whichever comes first, the employee must receive the current federal minimum wage or the state minimum wage, whichever is higher. | Employees must meet certain criteria to receive the minimum wage. | The minimum wage is enforced by law; The minimum wage applies broadly | |
| AFS | [Topic: death penalty] If we don't stop the death penalty now, we will put innocent people to death while we try to fix the problems. | The death penalty puts innocent people to death; The death penalty needs to be put on hold while it is being fixed | The implementation of the death penalty is flawed | The death penalty should be stopped now |
| SemEval16 | [Topic: feminist movement] Does it involve women? It's sexist. Does it not involve women? It's sexist. No winning. | There is no winning against accusations of sexism | Accusations of sexism are indiscriminate | Feminist arguments are inconsistent |

Table 8: Exemplars for inferential decomposition for the source type of argument/stance datasets. We sample $n$ exemplars from this set to form a prompt, per Table 6.

| Utterance | Propositions: Explicit [Required] | Inferences about Utterance Subject [Optional] | Inferences about Utterance Perspective [Optional] |
|---|---|---|---|
| Carlos Delfino got cookies and dipped em on Durant's head | Carlos Delfino dunked on Kevin Durant | Kevin Durant played against Carlos Delfino in basketball | |
| Real Madrid out to kill lewandowski | Real Madrid is trying to injure Lewandowski | Injuring a key player will help Real Madrid win
Lewandowski is an important player | |
| Zach Ertz taken by the damn Eagles | The Philadelphia Eagles drafted Zach Ertz | Zach Ertz is a football player | The Eagles are a hated team |
| Emotionally I can't handle when Mufasa dies | It is emotional when Mufasa dies | Mufasa dies in the Lion King | The Lion King is an emotional movie
Mufasa's death is sad |
| Roy Nelson just smoked Kongo | Roy Nelson knocked out Kongo | Roy Nelson and Kongo were in a combat sport | |
| Russell Westbrook getting knee surgery | Russell Westbrook will recieve knee surgery | Russell Westbrook injured his knee | |
| Boston Marathon Bombers planned to target Times Square | Boston Marathon Bombers planned to attack Time Square, NYC | The Boston Marathon bombers were apprehended
Terrorists bombed the Boston Marathon | |

Table 9: Exemplars for inferential decomposition for Twitter-PC. We sample $n$ exemplars from this set to form a prompt, per Table 6.

| Utterance | Propositions: Explicit [Required] | Inferences about Utterance Subject [Optional] | Inferences about Utterance Perspective [Optional] |
|---|---|---|---|
| Stop illegally forcing the clot shot onto the citizens and their children. This is wrong and you are taking away our freedom. The Covid vaccine is killing people. | The covid vaccine kills people. | The covid vaccine causes blood clots.

The government is illegally forcing the covid vaccine on people. | Forcing the covid vaccine limits freedom. |
| Kids don t get the fake coved flu and then vaccines don t work if you can still gat it. Refer to Colin Powell | Covid is not real.

Children do not contract coronavirus | Covid vaccines do not prevent covid. | |
| Bodily autonomy is everyone s right and extremely important. everyone should have the right to choose whether or not they want to have a vaccine. It should not be mandated for any profession or for children at any level of education as it violates their rights. | Vaccine mandates violate bodily autonomy. | Vaccine mandates violates the rights of citizens. | People should choose whether they get vaccinated. |
| It appears to me that there is a lack of fidelity in this entire VACCINE process, deemed scientific . There are SO many inconsistencies from the onset of how this vaccine was going to stop covid. From my stand point, half of those I know now have health issues after their shot s are unable to function in their daily routines. I know health professionals , that when asked if they were submitting VAERS info, the reply is We simply do not have the time or Have you ever tried to input Vaers info? It s IMPOSSIBLE!! | The covid vaccine causes serious health problems. | Health professionals don't report adverse effects of vaccines.

The covid vaccine approval process is unscientific. | There may be a conspiracy to limit information about vaccine side effects. |

Table 10: Part one of the exemplars for inferential decomposition for FDA comments. We sample $n$ exemplars from this set to form a prompt, per Table 6. The other exemplars for FDA comments are provided in Table 11.

| Utterance | Propositions: Explicit [Required] | Inferences about Utterance Subject [Optional] | Inferences about Utterance Perspective [Optional] |
|---|---|---|---|
| Do not force children to take a dangerous vaccine for a sickness that is nearly nonexistent in that age range and is easily treatable with other medications. | Covid is nearly nonexistent in children. Covid is treatable with other medications. The covid vaccine is dangerous. | The government wants to force people to vaccinate. | |
| Our children need to be protected from experimental vaccines. The proper protocol has not been followed and our children should not be guinea pigs and put it risk. See the Nuremberg trials, you will be held accountable. | The covid vaccine is experimental. Those mandating the vaccine will be held accountable. The proper protocol to approve vaccines was not followed. | The use of covid vaccines in children is a human rights violation. | People promoting COVID vaccines are acting like Nazis |
| Please do not offer vaccinations for kids 5 11. They have beautiful immune systems to keep them healthy and fight off viruses and bacteria. | Young children have strong immune systems. | Children are not susceptible to complications from covid. | Children are more robust to illness than adults |

Table 11: Part two of the exemplars for inferential decomposition for FDA comments. We sample $n$ exemplars from this set to form a prompt, per Table 6. The other exemplars for FDA comments are provided in Table 10.

| Utterance | Propositions: Explicit [Required] | Inferences about Utterance Subject [Optional] | Inferences about Utterance Perspective [Optional] |
|---|---|---|---|
| The #HonestAds Act will strengthen protections against foreign interference in our election. No more election ads paid for in rubles. | The Honest Ads Act will strengthen protections against foreign election interference | Russia will be prevented from purchasing election advertisements; Russia interfered in the 2016 presidential election | Foreign interference in US elections is wrong |
| Our nation is hurting. George Floyd's death was horrific and justice must be served. A single act of violence at the hands of an officer is one too many. George Floyd deserved better. All black Americans do. Indeed, all Americans do. | A police officer killed George Floyd | George Floyd's death was unjust | Black Americans deserve better treatment by police; Police should not be violent without cause; The institution of policing has flaws |
| Happy Wyoming Day! Today, our great Equality State celebrates 151 years of being the first to officially recognize women's inherent right to vote and to hold office. | Wyoming was the fist state to recognize womens' right to vote | Wyoming supports gender equality | Womens' rights should be supported |
| "More apologies from Mark Zuckerberg won't fix Facebook. We need accountability and action — not vague commitments to do better while continuing to profit off of users' personal data. | Mark Zuckerberg makes insincere apologies; Facebook is trying to avoid accountability | Facebook is not committed to user privacy; Tech companies profit off users' personal data | Personal data of users should be protected |
| Finding a permanent solution to #ProtectDreamers is as urgent a task as ever. President Trump created this crisis, and he should stop tanking bipartisan congressional efforts to solve it. We owe it to these kids to keep them in the only country they've ever known as home. | There is an urgent need to help DACA recipients | DACA receipients should recieve amnesty; People who immigrated as children are rightful citizens | Illegal immigrants should be treated humanely; Donald Trump is anti-immigrant |
| Qualified immunity reform should have as its focus professionalizing police departments, institutionalizing best police practices when it comes to use of force, and protecting constitutional rights of American citizens. | Qualified immunity reform should focus on formalizing police behavior; Rules around police use-of-force need to be codified | Qualified immunity is in need of reform; Police sometimes violate Americans' constitutional rights | Police perform an important function in society |
| Survivors of the coronavirus show symptoms of ME/CFS, a debilitating and chronic illness that already impacts 2.5 million Americans. I am fighting to secure the funding needed to treat this disease and give patients the care they need. | Many Americans are battling the long term effects of Coronavirus; Treating patients with Chronic Fatigue Syndrome requires funding | Coronavirus can have adverse long-term effects | People affected with Long Covid deserve treatment |

Table 12: Exemplars for inferential decomposition of legislative tweets, used in the covote prediction task in Section 5. We sample $n$ exemplars from this set to form a prompt, per Table 6.

## Introduction

In this survey, you will be answering simple questions about written content from different sources, like news articles and social media posts.

## Instructions

You will review **two pieces of written content** and will answer two questions about the relationship between them. We call these pieces of writing Text 1 and Text 2.

1. You will answer how **reasonable it is to conclude that someone who said the *first text* would also say the *second text***. In some cases, Text 1 may be expressing an opinion or belief. You should answer this question based on what the person who said Text 1 would say (not necessarily what you think). Sometimes, Text 1 may be factual, like a news story. Here, you should answer whether Text 2 is plausibly *true* based on Text 1.

2. You will answer whether Text 2 **adds new information** to Text 1 or **changes the information** in Text 1. By new information, we mean something that *was not explicitly said* in the Text 1. A paraphrase does not add new information.

The answer may not always be obvious, so use your best judgment.

Figure 8: Instructions used in the survey for human validation of the generations as discussed in § 3.

## Examples

Text 1: *There is a teal teapot in the kitchen.*
Text 2: *The kitchen contains a teal teapot.*
Answer: If someone said Text 1, then it is **definitely reasonable** that they would also say Text 2. Text 2 **does not add new information** to Text 1.

Text 1: *The cat meowed in the kitchen before its usual dinnertime.*
Text 2: *The cat was hungry.*
Answer: If someone said Text 1, then it is **probably reasonable** to conclude that they would also say Text 2. Text 2 **adds new information** to Text 1.

Text 1: *I'm really hungry right now!*
Text 2: *I just ate a big meal.*
Answer: If someone said Text 1, then it is **probably not reasonable** to conclude that they would also say Text 2. Text 2 **adds new information** to Text 1.

Text 1: *The plandemic is a massive attempt to place global power into a one dictatorship based on a complete lie!*
Text 2: *COVID is a hoax*
Answer: If someone said Text 1, then it is **probably reasonable** to conclude that they would also say Text 2. Text 2 also **adds new information** to Text 1.

Text 1: *The arraignment of former President Donald Trump has concluded. Trump pleaded not guilty to 37 charges related to alleged mishandling of classified documents.*
Text 2: *Donald Trump pleaded not guilty to charges.*
Answer: If someone said Text 1, then it is **definitely reasonable** to conclude that they would also say Text 2. Text 2 **does not add new information** to Text 1.

Text 1: *Barack Obama was elected president in November 2016.*
Text 2: *Barack Obama was elected president in May 1900.*
Answer: If someone said Text 1, then it is **definitely not reasonable** to conclude that they would also say Text 2. Text 2 **adds new information** to Text 1.

Figure 9: Examples used in the survey for human validation of the generations as discussed in § 3.

## Introduction

In this survey, you will be answering questions about small collections of comments. All comments in this survey are related to **COVID-19 vaccines in children.**

As background, the FDA approved vaccination for children ages 5-11 in 2021. During this approval process, the FDA sought public commentary. You will be reading excerpts of that commentary.

**Note that much of the commentary makes reference to known misinformation, falsehoods, and unsubstantiated claims. You should not take the commentary at face value!**

## Instructions

You will review short lists of commentary, then answer a few questions about **how related** the items are in each list. Everything is related to COVID vaccination in children, so you are interested in what makes them related *other* than that fact.

First, you will read the collection and **write a label for what they have in common**. A label can be one or more words that succinctly describe the collection. Try your best to write a descriptive label.

Then you will answer **how related** the collection is, *other* than the fact that they are all about COVID vaccination. The scale is from **1 - Very loosely related** to **5 - Very closely related**.

Finally, you will read two more documents, and **select the one you think best belongs** with the others.

Figure 10: Instructions used in the survey for human evaluation of clustering quality as discussed in § 4.

## Examples

Here, we will show two examples of comment collections that all relate to the *economy*.

### Closely-related Collection

1. Its profiteering, not inflation. Inflation is the term they use to cloak the reality and make you think no one is causing this or is responsible. Its profiteering.
2. A lack of antitrust enforcement is the main driver because it allows companies to raise prices without consequences from competitors or the government.
3. The greed at the corporate level is getting out of control. Not sure how long they expect to able to do this, how much they think they can do to make their employees' lives excruciatingly difficult, and things too expensive for consumers, all the while they're taking in obscene salaries.
4. Inflation is happening, and is caused by many economical factors. However it is being exasperated by corporate greed, inflation is just the mask.

These comments are all **very closely related**. A descriptive label might be **"inflation and corporate greed"**, because the comments discuss corporations raising prices to increase profits.

### Loosely-related collection

1. Ban all cryptocurrency entirely and get all of the functional benefits by issuing a central bank digital currency instead.
2. The US is already one of the most economically diverse, least trade dependent nations with a wide variety of secure supply lines.
3. All I can say is that I recently noticed that the ONE THING I still ~~splurge~~ splurgED on (Clif bars - they're my husbands favorite and he has one for breakfast every day) have gone up nearly $1.50 in 4 months.
4. I only apply to jobs that have the salary listed now because inevitably if it's not listed, it's a terrible salary. I'm so thankful that states are passing salary transparency laws that require companies to post the pay range in job postings.

These comments are all **very loosely related**. It is hard to think of a more descriptive label than something general like **"the economy"**.

Figure 11: Examples used in the survey for human evaluation of clustering quality as discussed in § 4.