# OpenReview forum: "Natural Language Decompositions of Implicit Content Enable Better Text Representations"
_EMNLP/2023/Conference — EMNLP 2023 Main_

### Official Review · Reviewer_V5xX · 2023-07-21

**Typos Grammar Style And Presentation Improvements:** see Questions
**Soundness:** 3

**Excitement:**

3: Ambivalent: It has merits (e.g., it reports state-of-the-art results, the idea is nice), but there are key weaknesses (e.g., it describes incremental work), and it can significantly benefit from another round of revision. However, I won't object to accepting it if my co-reviewers champion it.

**Paper Topic And Main Contributions:**

Content: The paper describes a technique to text analysis by formulating (using LLM’s) propositions that contain more explicit knowledge beyond the implicit meaning of a certain utterance. The idea behind is that the intention of an utterance is not necessarily 100% visible in a certain expression (formulation), hence, extra (world) knowledge is helpful to improve text analysis at scale. The technique is tested against a crowd-souring approach, and two downstream tasks (clustering comments on one controverse topic, and legislator “similarity-based prediction”).

**Questions For The Authors:**

Question 1: Regarding operationalization: The proportions are formulated following instructions from the appendix, and these examples are prompted to the LLM together with the instructions. An example of these instruction in the paper would suffice. The appendix should contain material to support the paper, but is no substitute to extra pages missing. Main content to present the work in a clear manner should be in the main paper (1st eight pages).

Question 2: I cannot follow well the break between the formula in L 137 and the one in L 140. I understand the rationale behind the formula in L 137 (formula numbering makes communication easier), but I don’t understand why it is presented, when the MI between the intension and relevant information-enriched expression is not calculated, and instead inferred propositions are requested via the LLM. How do both formulas relate to each other?

Question 3: L 169: Figure 1 is not the same examples as the one in the text. The reference here is confusing.

Question 4: L 252: You calculate the similarity between the original sentences? What is the relation between to sentences? Based on what are two sentences selected? Is it any two sentences?

Question 5: L 259: Correlation or similarity?

Question 6: L 331: Are the two authors you mention, authors from the paper? 31 examples refer to one comment or all comments?

Question 7: L 448: Please introduce beta_0 and beta_k.

Question 8: L 480: Why do you calculate similarity between two legislators? What happens next? What do the two sets of similarities regard to?

Question 9: L 510: Figure 4 is difficult to read. Is this an embeddings dimension reduction and a clustering visualization in one figure? I did not find an explanation on the parameter details of this figure.

**Reasons To Accept:**

Positive: I enjoyed the motivation and overall idea to overcome the gap between intention and expression. I really think that this is a very important niche and improving/developing supportive, automated ways to analyze communication in any social field is really an underestimated task in our society. I also see some merits in the overall design and approach to make use of LLM prompting to collect extra world knowledge.

**Reasons To Reject:**

Problems: I understand, you operationalize your study by clustering and measuring embeddings similarities of your augmented data. This way you can explain the results more closely. However, I feel a bit lost in the way you design the experiments. A better structuring of the work can help the reading. When you explain the setup of an experiment, give structured details on the Research Question, the data and its format, the technique you use, the parameters for the algorithm and an explanation of the parameter values selection. You often refer to the literature, which is good, but give to few details on the technology you use. Often similarities are calculated, but these are not justified well and nor are they properly referred to (“do not let your appendix speak for you”).

**Reproducibility:**

2: Would be hard pressed to reproduce the results. The contribution depends on data that are simply not available outside the author's institution or consortium; not enough details are provided.

**Reviewer Confidence:**

4: Quite sure. I tried to check the important points carefully. It's unlikely, though conceivable, that I missed something that should affect my ratings.

---

> ### Author Rebuttal · Authors · 2023-08-28
>
> Thank you very much for your review. We are very pleased that you believe this work fills an important but understudied niche, and that it addresses the key problem of automatically analyzing social communication by relying on implicit information. We are also glad that you find merits in our motivation and the design of our approach. We also particularly appreciate your constructive feedback on the presentation of our experimentation.
>
> It appears that your concerns revolve primarily around the way we presented the experimentation, and the division of material between the main body and the appendix, not methodological soundness *per se*. As discussed in our response to Reviewer 1, we will expand on the method (e.g., prompt instructions) in the main text. We will also update the experimental sections according to your suggestions:
> * We will put the research questions at the beginning of each section:
>     * Section 4 (Method validation):
>         * Do language models reliably produce plausible explicit & implicit propositions (as judged by human annotators)?
>         * Under the assumption that human similarity judgments make use of both explicit and implicit information, does including such  information in sentence representations improve automated estimates of similarity?
>     * Section 5 (Analysis of public commentary):
>         * Does the representation of comments’ explicit & implicit content lead to improved discovery of themes in a corpus of public opinion (as judged by human annotators)?
>     * Section 6 (Analysis of legislative behavior):
>         * Does the automated similarity between legislators’ explicit & implicit propositions help explain the similarity of their voting behavior?
> * We will explain key hyperparameters & datasets at the beginning of each section, and justifications for our decisions where they do not already exist (as you note, many are included in the references)
> * We will clarify the differences between the similarities used:
>     * In Section 4, we will emphasize that similarities are computed over pairs of sentences from existing semantic textual similarity (STS) datasets
>     * In Section 6, we will move the details of the legislator similarity computation from the appendix to the main text. Similarities are computed over two legislators, who produce multiple tweets, and therefore we must aggregate over all sets of tweets.
>
> **[Responses to questions]**
> * Q1: As discussed, we will illustrate instructions in the main paper, rather than in the appendix.
> * Q2: We appreciate your pointing out this lack of clarity and will update the explanation in the final version (and number the equations!).
>     * As you note, our goal here is to build intuition, and it does not influence our method. We want to show that, over sampled enriched / augmented expressions $R$ from the LLM (equation on L140), the mutual information between intent and these augmented expressions (LHS of equation on L135) increases relative to the MI between the intent and original expression alone. That is, $\text{I}(E, R ; I ) > \text{I}(E ; I)$. The identity on L135 shows that this is the case, assuming $\text{I}(R; I | E) > 0$. Hence, we do not need to calculate it directly: we illustrate that by sampling from an LLM, this MI increases, as desired.
> * Q3: Figure 1 contains an actual, real-world example, whereas the text contains an illustrative example to help with explanation. We can make the two the same in the final version.
> * Q4: This section applies our approach to the standard semantic textual similarity (STS) task (e.g., Cer et al. 2017). Sentence pairs come from existing STS and argument similarity datasets, introduced on lines 204-205 and 209.
>     * Each dataset consists of $n$ sentence pairs, and annotators score each pair based on the semantic similarity. We follow prior work and construct distributed representations (i.e., embeddings) of each sentence in a pair, then compute an automated similarity (here, cosine).  So “I like NLP” / “I love NLP” might receive a human score of 4.5 out of 5, and the cosine similarity of the embeddings might be 0.9
> * Q5: This answer follows from Q4. We compute the spearman correlation between embedding similarities and human-annotated similarities across all sentence pairs in each dataset (Table 1). The higher correlation implies that embeddings based on our method are more aligned with human similarity judgments.
> * Q6: Yes, the exemplars are created by two of this paper’s authors following a consensus process. Multiple propositions are produced for 7 original comments, totalling 31 (comment, proposition) pairs; we will make this explicit in the final version.
> * Q7: The $\beta$ are learned parameters of the regression model, we will introduce them right after the equation. Thank you for pointing this out.
> * Q8: We can include details of the similarity calculation in the main text, instead of the appendix.
>     * Prior work computes similarities between legislators to help explain their joint voting behavior, and we follow this experimental setup (see references on lines 407-419). We measure the embedding similarity between legislators using different representations of their online speech. Specifically, we compare similarities based on their observed tweets (a baseline) to those based on generated propositions (our proposed method). Our hypothesis is that legislators whose speeches contain semantically similar propositional content will have similar voting behavior. We find evidence supporting this hypothesis.
> * Q9: Yes, the graph is representing both a dimension reduction of the embeddings as well as a k-means clustering of those embeddings ($K=10$). Two clusters are circled, which correspond to Republican/conservative and Democratic/liberal framings of the topic of abortion. We will elaborate on the hyperparameters and the method used to generate this figure, and refine it to be more straightforward.
>
> **[Reproducibility]** We recognize this is somewhat subjective, but as noted to reviewer 2, all datasets are public, our appendices include all prompts and exemplars, and code was provided with the supplementary materials (which we will release on acceptance). With regard to human participation in our experimentation, we will also release the human data if accepted, and detailed human annotation instructions are provided in the Appendix.
>
> **References:**
> * SemEval-2017 Task 1: Semantic Textual Similarity Multilingual and Crosslingual Focused Evaluation (Cer et al., SemEval 2017)

---

### Official Review · Reviewer_5xdd · 2023-08-04

**Soundness:** 3

**Excitement:**

3: Ambivalent: It has merits (e.g., it reports state-of-the-art results, the idea is nice), but there are key weaknesses (e.g., it describes incremental work), and it can significantly benefit from another round of revision. However, I won't object to accepting it if my co-reviewers champion it.

**Paper Topic And Main Contributions:**

This paper proposes a method for generating implicitly related propositions of an observed text using large language models.
A prompt is designed to make large language models output texts that describe propositions behind the input text.
This method is evaluated by human evaluation, content clustering, and predicting legislative behaviors.
The style of this paper is not like a typical NLP paper, which mainly discusses technological ideas, but is more focused on the problem and task settings in the context of social science.

**Reasons To Accept:**

- It is interesting to see LLMs are effectively applied to problems in social sciences. The discussion on the importance of these tasks is valuable for the NLP community.

**Reasons To Reject:**

- The proposed method reminds me of traditional methods for extracting common sense knowledge (like script knowledge, causal relations, etc.), while their relationship is not discussed. Similar ideas have been discussed repeatedly in the history and explaining the essential differences from such previous ideas should be necessary.
- No qualitative analysis of the proposed method is given. As an NLP perspective, it is important to investigate why the proposed method increases performance.
- It is not clear what "interpretation at scale" means. The title should describe the main content of the paper properly.

**Reproducibility:**

3: Could reproduce the results with some difficulty. The settings of parameters are underspecified or subjectively determined; the training/evaluation data are not widely available.

**Reviewer Confidence:**

3: Pretty sure, but there's a chance I missed something. Although I have a good feel for this area in general, I did not carefully check the paper's details, e.g., the math, experimental design, or novelty.

---

> ### Author Rebuttal · Authors · 2023-08-28
>
> Thank you for noting that our work is an effective application of LLMs to problems in social sciences, and for viewing our work and discussion as valuable for the NLP community.
>
> **[Commonsense knowledge]** We agree that our inferential decomposition method relates to commonsense knowledge extraction. In our related work section (lines 532-538) we discuss some prior work more specifically focused on commonsense inference. If accepted, we will add relevant references to LLMs’ capacity for commonsense extraction (e.g., Trinh and Le 2018; Bosselut et al. 2019; Yang et al. 2020; Bian et al. 2023), as well as traditional methods like script induction (e.g., Weber et al. 2020) and template-based approaches (e.g., Tandon et al. 2014). These are important precursors to our work. At the same time, we emphasize that our decompositions are not constrained to commonsense knowledge alone.
>
> **[Qualitative Analyses]** We agree that qualitative analyses are important.  We qualitatively examined our method in an intrinsic way: in Table 3, we show that crowdworker-assigned labels of our method’s clusters match narrative themes discovered by experts in prior work; meanwhile, none of the crowdworker labels for baselines recover those themes (Lines 387-391). In Figure 4, we provide examples showing that clusters based on our method reveal opinions at a high level of granularity.  Regarding downstream tasks, in our sentence-similarity experiments (Table 1), we found our method improves correlations with human judgments of Tweet similarity, but does not perform as well on SICK-R (which uses image/video captions). In an unreported experiment, two authors independently scored a sample of 100 LLM-generated paraphrases from the Twitter-STS and SICK-R datasets, and found that LLMs often inject implicit text for Twitter (16%) but never for SICK-R (significant at p<0.05). This led us to conclude that implicit text is important when evaluating the similarity of social language, which informs our CSS experiments in Sections 4 and 5. An example of how our inferential decomposition method can place two tweets closer in embedding space is given in Figure 1.
>
> We will contextualize this discussion in terms of qualitative performance analysis if accepted; specifically, we will include examples where legislator similarities based on our method better explain voting similarity than the baseline.
>
> **[Meaning of “scale”]** We appreciate your noting that our use of the term “scale” was not clear. Repeating our response to Reviewer 1, we use our method to automatically “interpret” the implicit & explicit propositional content in large amounts of text data, which we in turn aggregate to help the end-user identify broader themes (in Section 4). In this way, it functions similarly to a topic model, which also facilitates interpretation of text corpora. Hence, the method facilitates efficient analysis of large-scale datasets (e.g., over 5,000 documents). In addition, we note that our method significantly improves reading times of individual text items (see “Evaluation Time” in Figure 3), facilitating human engagement with large datasets. We will clarify this in the final version to make the relationship with the title more evident.
>
> **[Reproducibility]** All datasets are public, our appendices include all prompts and exemplars, and code was provided with the supplementary materials (which we will release on acceptance). With regard to human participation in our experimentation, we will also release the human data if accepted, and detailed human annotation instructions are provided in the Appendix.
>
>
> **References:**
> * Acquiring Comparative Commonsense Knowledge from the Web. (Tandon et al., AAAI 2014).
> * A Simple Method for Commonsense Reasoning. (Trinh and Le., arxiv 2018)
> * COMET: Commonsense Transformers for Automatic Knowledge Graph Construction (Bosselut et al., ACL 2019)
> * Designing Templates for Eliciting Commonsense Knowledge from Pretrained Sequence-to-Sequence Models (Yang et al., COLING 2020)
> * Causal Inference of Script Knowledge (Weber et al., EMNLP 2020)
> * ChatGPT is a Knowledgeable but Inexperienced Solver: An Investigation of Commonsense Problem in Large Language Models. (Bian et al., arxiv 2023)

---

### Official Review · Reviewer_xCQB · 2023-08-07

**Soundness:** 3

**Excitement:**

4: Strong: This paper deepens the understanding of some phenomenon or lowers the barriers to an existing research direction.

**Paper Topic And Main Contributions:**

This paper presents a method that captures not only the information explicitly encoded in a text unit, but also the associated knowledge that a person relies on when understanding a text. The goal is to support machine learning tasks that rely on interpretation, such as many tasks in computational social science, through datasets with enriched knowledge. Instead of using human annotators to generate the so-called inferential decompositions, the method proposes to use LLMs and then manually validate the generated data for reasonableness and whether it contains new information not explicitly included in the original text unit. Experiments prove the usefulness of the method in two practical use case scenarios: e-rulemaking and legislator behavior. It shows that inferential decompositions lead to a better, more compact and easier to interpret clustering (thematically, opinions).

**Questions For The Authors:**

- A: Is step 4 of the method (line 156-157) intended to take place in practical use cases? And how high is the cost associated with this step?
- B: What do you mean by the term "scale" in your setup and title? There is no discussion in the paper so far.

**Reasons To Accept:**

- This approach appears to be a valuable method for incorporating implicit knowledge.
- A thorough assessment of the method's effectiveness is conducted, both in an intrinsical and extrinsical way via well-motivated use cases.

**Reasons To Reject:**

-  The proposed method saves manual annotation effort when creating the inferential decompositions. Afterwards, however, these are manually checked for reasonableness and novelty (step 4, line 156-157). It remains unclear how high this additional effort is, with which costs the application of this methodology is associated overall and whether this results in a serious limitation for the application. Or is this step of intrinsic validation only exemplary in the context of this work and omitted in practical application? (This would also fit with Figure 4: "our approach is also entirely unsupervised"). This point needs to be clarified and discussed.
- While the analysis and discussion is very detailed, the explanation of the method itself is very brief and can rather be found in the appendix. More detail is needed here in the main body.

**Reproducibility:**

3: Could reproduce the results with some difficulty. The settings of parameters are underspecified or subjectively determined; the training/evaluation data are not widely available.

**Reviewer Confidence:**

3: Pretty sure, but there's a chance I missed something. Although I have a good feel for this area in general, I did not carefully check the paper's details, e.g., the math, experimental design, or novelty.

**Typos Grammar Style And Presentation Improvements:**

- Typos, please check lines 173, 204-205 (missing commas)
- A variety of tables and figures are used in this work. This is somewhat cluttered, especially since the Tables/Figures are often not found on the page where they are referenced.
- If tables are referenced in the appendix, this should be made clear (e.g., see Table 11 in Appendix A.x).
- The references have to be checked, sometimes page numbers are missing.
- Figure 3: What are “generation clusters”?

---

> ### Author Rebuttal · Authors · 2023-08-28
>
> Thank you for the feedback, and for recognizing our method as a valuable way to incorporate implicit knowledge in computational social science applications. We also appreciate the acknowledgement of our thorough method assessment for two real-world use cases.
>
> **[Validation of Decompositions]** We’ll respond both to your first “reason to reject” and question A. Step 4 of our protocol involves the validation of LM-generated outputs for a sample from the target dataset. Yes, it is not strictly necessary, because we validate outputs across five datasets spanning diverse domains. In practice, we would recommend this validation check for specialized datasets or sensitive use cases. Even in such cases, this step is inexpensive: a sample of 150 annotations are sufficient for a 5% margin of error when estimating the proportion of valid generations (95% CI, assuming a lower-bound of $p=0.85$, as found in this work). This takes an end-user 50 minutes (based on the median crowdworker time of 20s/item), or 37.5 USD for 3 crowdworkers per item @ 15 USD/hour.
>
> Thus, this is a relatively low cost optional step, and the method is indeed fully unsupervised if it is skipped. Our main goal was to follow best practices in computational social science, which is to validate unsupervised model outputs, e.g., topic models (see Grimmer & Stewart, 2013). We agree that this was not clearly distinguished in our paper, and we will address this in the writing of the final version.
>
> **[Method details]** We also agree with your second point that too much of the method is in the appendix, and will move key details to the main text (e.g., discussing a relevant example of the prompt instructions).
>
> **[Meaning of “scale”]** Question B: Thank you for pointing out that we were not sufficiently clear with our usage of the term “scale”. We use our method to automatically “interpret” the implicit & explicit propositional content in large amounts of text data, which we in turn aggregate to help the end-user identify broader themes (in Section 4). In this way, it functions similarly to a topic model, which also facilitates interpretation of text corpora. Hence, the method facilitates efficient analysis of large-scale datasets (e.g., over 5,000 documents), with an optional amount of validation involving at most a small subset, as discussed above. In addition, we note that our method significantly improves reading times of individual text items (see “Evaluation Time” in Figure 3), facilitating human engagement with large datasets. We will clarify this in the final version to make the relationship with the title more evident.
>
> We will make the suggested presentation improvements. We will also correct the term “generation clusters”, which should have read “*decomposition* clusters”.
>
> **References:**
> * Grimmer, Justin, and Brandon M. Stewart. "Text as data: The promise and pitfalls of automatic content analysis methods for political texts." Political analysis 21, no. 3 (2013): 267-297.

---

### Meta-Review · Area_Chair_xp5e · 2023-09-12

**Recommendation:** 5

**Metareview:**

This paper presents a method for the “interpretation” of text at scale. The authors propose using LLMs to generate propositions which are “inferentially related” to an original text (see AC suggestions at the bottom). They propose that these propositions provide “explicit representations” of the “implicit” interpretations that come with reading textual information. The authors validate generated propositions with a simple crowd experiment, and test the utility of generated propositions for clustering public comments and predicting legislator voting patterns.

Reviewers offered a largely positive assessment of the soundness and excitement of this work, especially after discussing with authors during the rebuttal period. All soundness and excitement scores were 3 or higher (following discussion). Reviewer xCQB described the paper as “a valuable method for incorporating implicit knowledge” and suggested that the authors provided a “thorough assessment of the method's effectiveness.“ Reviewer 5xdd noted that it was interesting to see how LLMs could be “effectively applied to problems in social sciences” and reviewer V5xX liked that it supported “automated ways to analyze communication” in a social field.

However, reviewers did offer some negative comments about the paper. The AC has summarized negative comments below, in order of what the AC believes to be the most and least important to acceptance/rejection decisions.

Reviewer 5xdd noted a lack of comparison with “traditional methods for extracting common sense knowledge.” In the AC’s opinion, this does seem like a limitation of the work. But this limitation should be placed in a broader context. Overall the authors did a good job grounding their submission in recent work from NLP and CSS, so this shortcoming does not seem core to the underlying soundness or excitement of the submission.

- Note: in the discussion, the authors replied by suggesting that “commonsense knowledge” and “inferentially related” knowledge are not equivalent. They included references to work on LLMs for commonsense reasoning. But some of the nuances between inferentially-related reasoning and commonsense reasoning did not come through on OpenReview.  See the AC’s suggestion about “inferentially related” at the end of this review.

Reviewer xCQB wondered if the method required human validation, which might limit the utility of the work. The authors replied that human validation was a “low cost optional step” best applied in “sensitive” use cases. Figure 2 seems to show that (at least for these experiments) human plausibility judgments seem to validate the LLM. Thus it seems like in the future you could use this method with less crowdworker validation (in some settings). So this does not seem like a major shortcoming of the work.

Reviewer 5xdd asked for qualitative analysis of the method, especially regarding cases where the method does not work. The authors replied with explanations of the ways in which their paper does incorporate qualitative analysis, and promised to contextualize the paper with examples upon acceptance. They also promised to analyze cases where “the method does not perform as well relative to the baseline.” Reviewer 5xdd replied to say that the authors’ “responses make sense to me mostly” which seems to largely address this issue, provided the authors make the requested changes.

Some of the reviews critiqued the style of the submission. These things might be changed during future revisions, and do not seem central to acceptance decisions. The requested stylistic changes are as follows. (Note: the AC agrees with reviewer 5xdd about the title.)
- Both reviewer xCQB reviewer V5xX noted that the authors should add more detail about the method in the main body of the paper. The authors agreed that this stylistic change was necessary during discussion. This is a stylistic issue that could be addressed during revision.
- Reviewer 5xdd pointed out that the title of the paper may not reflect its content. They noted that the paper should have a more concrete and descriptive title that accurately describes the body of the work.

While this metareview largely focuses on negative feedback, in total, it seems like the reviewers offered a largely positive assessment of the paper, and that the limitations they brought up appeared to be (1) addressed during the rebuttal period or (2) not central to the soundness or excitement of the work.

Suggestions from AC: It was hard to find the precise definition of “inferentially related” in this paper. The related work section helped explain some of the ideas behind the approach, but a more explicit statement of what “inferentially related” means would help the paper, even if the “formal semantic representation” (L525) of this phrase is itself “relaxed” (L535). Also, the AC agrees with 5xdd that the paper should have a more clear and descriptive title.

---

### Decision · Program_Chairs · 2023-10-07

**Decision:**

Accept-Main

**Comment:**

This paper presents a method for the “interpretation” of text at scale. The authors propose using LLMs to generate propositions which are “inferentially related” to an original text (see AC suggestions at the bottom). They propose that these propositions provide “explicit representations” of the “implicit” interpretations that come with reading textual information. The authors validate generated propositions with a simple crowd experiment, and test the utility of generated propositions for clustering public comments and predicting legislator voting patterns.

Reviewers offered a largely positive assessment of the soundness and excitement of this work, especially after discussing with authors during the rebuttal period. All soundness and excitement scores were 3 or higher (following discussion). Reviewer xCQB described the paper as “a valuable method for incorporating implicit knowledge” and suggested that the authors provided a “thorough assessment of the method's effectiveness.“ Reviewer 5xdd noted that it was interesting to see how LLMs could be “effectively applied to problems in social sciences” and reviewer V5xX liked that it supported “automated ways to analyze communication” in a social field.

However, reviewers did offer some negative comments about the paper. The AC has summarized negative comments below, in order of what the AC believes to be the most and least important to acceptance/rejection decisions.

Reviewer 5xdd noted a lack of comparison with “traditional methods for extracting common sense knowledge.” In the AC’s opinion, this does seem like a limitation of the work. But this limitation should be placed in a broader context. Overall the authors did a good job grounding their submission in recent work from NLP and CSS, so this shortcoming does not seem core to the underlying soundness or excitement of the submission.

- Note: in the discussion, the authors replied by suggesting that “commonsense knowledge” and “inferentially related” knowledge are not equivalent. They included references to work on LLMs for commonsense reasoning. But some of the nuances between inferentially-related reasoning and commonsense reasoning did not come through on OpenReview.  See the AC’s suggestion about “inferentially related” at the end of this review.

Reviewer xCQB wondered if the method required human validation, which might limit the utility of the work. The authors replied that human validation was a “low cost optional step” best applied in “sensitive” use cases. Figure 2 seems to show that (at least for these experiments) human plausibility judgments seem to validate the LLM. Thus it seems like in the future you could use this method with less crowdworker validation (in some settings). So this does not seem like a major shortcoming of the work.

Reviewer 5xdd asked for qualitative analysis of the method, especially regarding cases where the method does not work. The authors replied with explanations of the ways in which their paper does incorporate qualitative analysis, and promised to contextualize the paper with examples upon acceptance. They also promised to analyze cases where “the method does not perform as well relative to the baseline.” Reviewer 5xdd replied to say that the authors’ “responses make sense to me mostly” which seems to largely address this issue, provided the authors make the requested changes.

Some of the reviews critiqued the style of the submission. These things might be changed during future revisions, and do not seem central to acceptance decisions. The requested stylistic changes are as follows. (Note: the AC agrees with reviewer 5xdd about the title.)
- Both reviewer xCQB reviewer V5xX noted that the authors should add more detail about the method in the main body of the paper. The authors agreed that this stylistic change was necessary during discussion. This is a stylistic issue that could be addressed during revision.
- Reviewer 5xdd pointed out that the title of the paper may not reflect its content. They noted that the paper should have a more concrete and descriptive title that accurately describes the body of the work.

While this metareview largely focuses on negative feedback, in total, it seems like the reviewers offered a largely positive assessment of the paper, and that the limitations they brought up appeared to be (1) addressed during the rebuttal period or (2) not central to the soundness or excitement of the work.

Suggestions from AC: It was hard to find the precise definition of “inferentially related” in this paper. The related work section helped explain some of the ideas behind the approach, but a more explicit statement of what “inferentially related” means would help the paper, even if the “formal semantic representation” (L525) of this phrase is itself “relaxed” (L535). Also, the AC agrees with 5xdd that the paper should have a more clear and descriptive title.